

# 1 SAR Image Semantic Segmentation of Typical Oceanic and
# 2 Atmospheric Phenomena

Quankun Li[1,2,3], Xue Bai[1,2], Lizhen Hu[1,2], Liangsheng Li[4], Yaohui Bao[5], Xupu Geng[1,2,3], Xiao-Hai Yan[3,6]
[1] State Key Laboratory of Marine Environmental Science, College of Ocean and Earth Sciences, Xiamen University, Xiamen
361102, China
[2] Engineering Research Center of Ocean Remote Sensing Big Data, Fujian Province University, Xiamen 361102, China
[3] Joint Center for Ocean Remote Sensing, University of Delaware-Xiamen University, Xiamen 361005, China
[4] National Key Laboratory of Scattering and Radiation, Beijing 100854, China
[5] Fujian Hisea Digital Technology Co, Ltd, Sanming 365001, China
[6] College of Earth, Ocean & Environment, University of Delaware, Newark, DE 19716, USA
*Correspondence to*: Xupu Geng (gengxp@xmu.edu.cn), Xiao-Hai Yan (xiaohai@udel.edu)
**Abstract.** The ocean surface exhibits a variety of oceanic and atmospheric phenomena. Automatically detecting and
identifying these phenomena is crucial for understanding oceanic dynamics and ocean-atmosphere interactions. In this study,
we select 2,383 Sentinel-1 WV mode images and 2,628 IW mode sub-images to construct a semantic segmentation dataset that
includes 12 typical oceanic and atmospheric phenomena. Each phenomenon is represented by approximately 400 sub-images,
resulting in a total of 5,011 images. The images in this dataset have a resolution of 100 meters and dimensions of 256×256
pixels. We propose a modified Segformer model to segment semantically these multiple categories of oceanic and atmospheric
phenomena. Experimental results show that the modified Segformer model achieves an average Dice coefficient of 80.98%,
an average IoU of 70.32%, and an overall accuracy of 87.13%, demonstrating robust segmentation performance of typical
oceanic and atmospheric phenomena in SAR images.

## 21 1 Introduction

The exchange of energy and matter between the ocean and the atmosphere influences global water circulation, climate change,
and biogeochemical cycles. Its significant role in the global environment, climate, and ecological balance cannot be overstated.
Traditional ocean observations are predominantly based on in-situ or buoy observations. However, these methods incur high
observation costs and are limited in observation coverage, making it difficult to meet the demand for short-term and large-
scale ocean observations (Li et al., 2020). Compared to traditional methods, ocean remote sensing allows for distant, wide-
ranging, and efficient observation of the ocean. Synthetic Aperture Radar (SAR) is an active microwave remote sensing
imaging radar, characterized by all-daytime, all-weather, and high-resolution capabilities. Compared to optical satellites, SAR
can penetrate clouds, unaffected by weather conditions, making it especially advantageous for observing the ocean surface,
particularly in adverse weather conditions. Nowadays, the accumulation of a large number of SAR images has provided a
wealth of research data for ocean studies.

Traditional methods for detecting oceanic and atmospheric phenomena in SAR images primarily rely on feature selection and
threshold setting (Alpers and Huang, 2011; Chen et al., 2008; Fiscella et al., 2000; Topouzelis and Kitsiou, 2015). However,
these methods suffer from sensitivity to noise and poor generalization ability. With the application of artificial intelligence in
the field of oceanography, researchers have introduced deep learning methods, constructed data-driven models for detecting
oceanic and atmospheric phenomena, which extract features of different phenomena more accurately, significantly enhancing
the generalization ability of models.

Deep learning technology has demonstrated powerful image segmentation capabilities in the field of computer vision and has





become a reliable tool for extracting precise pixel objects in SAR images. Deep learning methods have been proposed to
automatically extract various ocean and atmospheric phenomena from SAR images, such as sea surface oil spills, sea ice,
ocean eddies, and internal ocean waves (Du et al., 2019; Zhang et al., 2021; Krestenitis et al., 2019; Zheng et al., 2022; Zi et
al., 2024). However, deep learning methods are data-driven method, and numerous studies have highlighted the challenges of
creating deep learning datasets, which require significant time and effort (Li et al., 2020). Additionally, most related research
has focused on limited areas or single phenomena, failing to achieve comprehensive observations of multiple oceanic and
atmospheric phenomena on the sea surface.

Fortunately, the first SAR image oceanic and atmospheric phenomena dataset for image classification was released by Wang
et al. (Wang et al., 2019b, a). This dataset manually selected Sentinel-1 WV mode images from 2016, annotated with 10 types
of geophysical phenomena: atmospheric fronts, biogenic slicks, icebergs, low wind speed areas, microwave convection cells,
ocean fronts, pure sea waves, rainfall cells, sea ice, and wind streaks. The results of the study indicated that deep learning
models based on this dataset achieved satisfactory results, with excellent classification performance. However, there are certain
limitations when multiple phenomena are present in SAR images. To address the situation where multiple phenomena exist in
a single image, Colin et al. compared various fully supervised and weakly supervised methods to segment different oceanic
and atmospheric phenomena at the pixel level (Colin et al., 2022). The experimental results showed that fully supervised
frameworks outperformed weakly supervised methods, effectively achieving the segmentation tasks for different oceanic and
atmospheric phenomena. Although Colin et al. achieved good fully supervised segmentation results using a U-Net-like
structure, we think that using only 100 manually annotated samples for each phenomenon is insufficient to achieve the best
segmentation results. Additionally, their study trained models with images at a resolution of 100 meters, but the output image
resolution was 400 meters, which could not capture fine structures and has certain limitations. Furthermore, although the study
demonstrated the potential for expanded applications of WV mode data, the dataset only included WV mode images, resulting
in limited data diversity, which is detrimental to semantic segmentation tasks.

Therefore, this paper aims to construct a SAR image dataset with multiple oceanic and atmospheric phenomena, annotated
manually, for fully supervised semantic segmentation tasks. By using a series of advanced semantic segmentation networks,
we aim to achieve pixel-level segmentation of various oceanic and atmospheric phenomena.

The paper is organized as follows: Section 2 describes the dataset construction; Section 3 describes the deep-learning model;
Section 4 presents the segmentation result and validation; Section 5 presents the cases analysis; and Section 6 shows the
summary and conclusion.
**2 Dataset**
**2.1 Focused phenomena**
Based on existing research and the classification of oceanic and atmospheric phenomena in SAR images in the TenGeoP-
SARwv dataset, we construct a SAR image semantic segmentation dataset comprising 12 types of oceanic and atmospheric
phenomena. The TenGeoP-SARwv dataset already includes 10 phenomena: Atmospheric Fronts (AF), Oceanic Fronts (OF),
Rainfall (RF), Icebergs (IC), Sea Ice (SI), Pure Ocean Waves (POW), Wind Streaks (WS), Low Wind Areas (LWA), Biological
Slicks (BS), and Micro Convective Cells (MCC). In addition to these, we have newly added two typical marine phenomena:
Oceanic Internal Waves (IWs) and Eddies (Eddy).

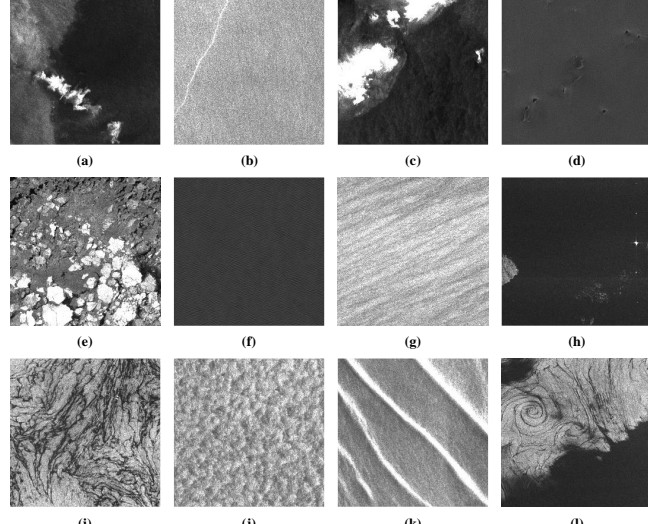

**Figure 1: 12 Oceanic and atmospheric phenomena we focus on (a) Atmospheric fronts; (b) Ocean fronts; (c) Rainfall; (d) Iceberg; (e) Sea ice; (f) Pure ocean wave; (g) Wind Strikes; (h) Low wind areas; (i) Biological slicks; (j) Micro convective cells; (k) Oceanic internal waves; (l) Eddy.**

## 2.2 Sentinel-1 Imagery Collection

Sentinel-1A and Sentinel-1B are launched by the European Space Agency (ESA) in April 2014 and April 2016, respectively. These polar-orbiting Earth observation satellites are primarily used for observing land and ocean through four imaging modes: Interferometric Wide swath (IW), Strip Map (SM), Extra Wide swath (EW), and Wave mode (WV). To enhance the applicability of the model and the diversity of the training data, we construct a semantic segmentation dataset of oceanic and atmospheric phenomena using images from the Sentinel-1 IW and WV modes. The IW mode is the primary acquisition mode for Sentinel-1 in land and coastal areas, while the WV mode is primarily used for open ocean areas.

For WV mode images, we incorporate the TenGeoP-SARwv dataset. Since this dataset is an image classification dataset with only one label per image, it cannot be directly used for pixel-level semantic segmentation tasks. Therefore, we select 2,383 WV mode images from the dataset for semantic segmentation annotation. Additionally, we reference the annotations proposed by Colin to further enhance the accuracy and reliability of the annotations.

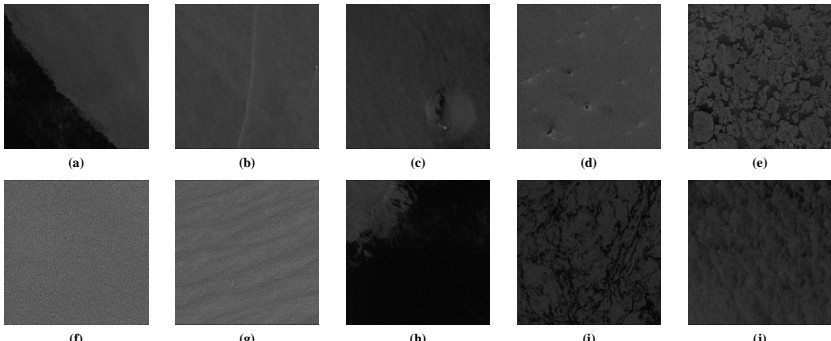

**Figure 2: TenGeoP-SARwv dataset image examples (a) Atmospheric fronts; (b) Ocean fronts; (c) Rainfall; (d) Iceberg; (e) Sea ice; (f) Pure ocean wave; (g) Wind Strikes; (h) Low wind areas; (i) Biological slicks; (j) Micro convective cells.**





For IW mode images, we select a total of 484 global Sentinel-1 IW mode images acquired from 2015 to 2022. We utilize the
Ground Range Detected (GRD) product of Sentinel-1 IW mode images. To maintain consistency in the processing of data
images, we apply a preprocessing method similar to that of TenGeoP-SARwv to the selected IW images (Wang et al., 2019b).
However, since IW mode images are often acquired near coastlines, we utilize a sea-land segmentation operation to eliminate
the influence of land. The dataset also includes the classification of artificial objects to mitigate the impact of vessels and
offshore wind turbines on the segmentation results of oceanic and atmospheric phenomena.

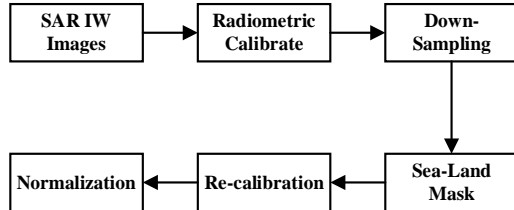

**Figure 3: Sentinel-1 IW mode images preprocessing method.**
Among these phenomena, due to the significant uncertainty in identifying oceanic internal waves, we reference the oceanic
internal wave object detection dataset proposed by Tao et al. (Tao et al., 2022a). However, the object detection task only
displays the object area without pixel-level annotation. From this dataset, we select 156 images with prominent ocean internal
wave features. Based on the object detection annotations from Tao's dataset, we determine the locations of oceanic internal
waves within the images to achieve accurate semantic segmentation annotations of ocean internal wave phenomena.

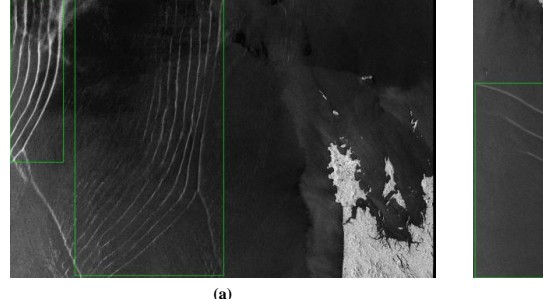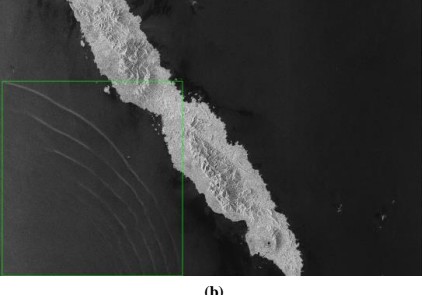

(a)                                              (b)

**Figure 4: The ocean internal wave object detection data set proposed by Tao et al.**

**2.3 Dataset Generation**
To construct a dataset suitable for semantic segmentation tasks, we use a sliding window method to crop the normalized 8-bit
and 16-bit images with a resolution of 100m into non-overlapping 256×256 sub-images. The 8-bit images are used for visual
interpretation and annotation, while the 16-bit images are used for model training. Each type of oceanic and atmospheric
phenomenon has approximately 400 sub-images. Then, based on the original SAR images, the Labelme software is used to
annotate the cropped sub-images. After annotation, the generated JSON files are batch-converted into PNG format files to
form the data labels. After obtaining image labels, all SAR image data from different categories and imaging modes are
randomly divided into training, validation, and test sets at a ratio of 8:1:1. This results in a total of 5,011 experimental data
samples, comprising 4036 training set images, 483 validation set images, and 492 test set images. The test set data is
independent of the training and validation sets.

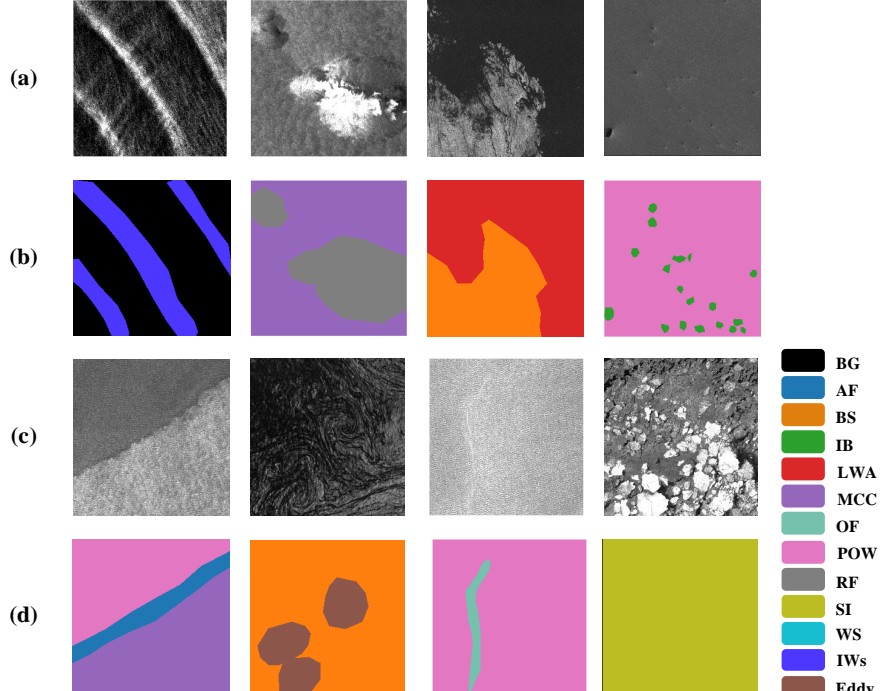

**Figure 5: SAR images and labels examples (a) (c) SAR Images; (b) (d) Image labels.**

## 3 Deep Learning Model

### 3.1 Modified Segformer

An improved Segformer model is employed for the semantic segmentation of marine and atmospheric phenomena in SAR images. While the original Segformer (Xie et al., 2021) possesses strong feature extraction capabilities and can generate multi-level feature maps, its decoder section employs simple MLP layers, which are ineffective in accurately restoring the detailed information from multi-scale feature maps, resulting in less precise segmentation outcomes. However, oceanic and atmospheric phenomena in SAR images exhibit multi-scale characteristics, complex features, and unclear boundaries. In this study, we enhance the original Segformer decoder by incorporating an improved Atrous Spatial Pyramid Pooling (ASPP) module (Chen et al., 2017), Coordinate Attention (CA) module (Hou et al., 2021), and employing a progressive upsampling approach to fuse feature maps of different scales.



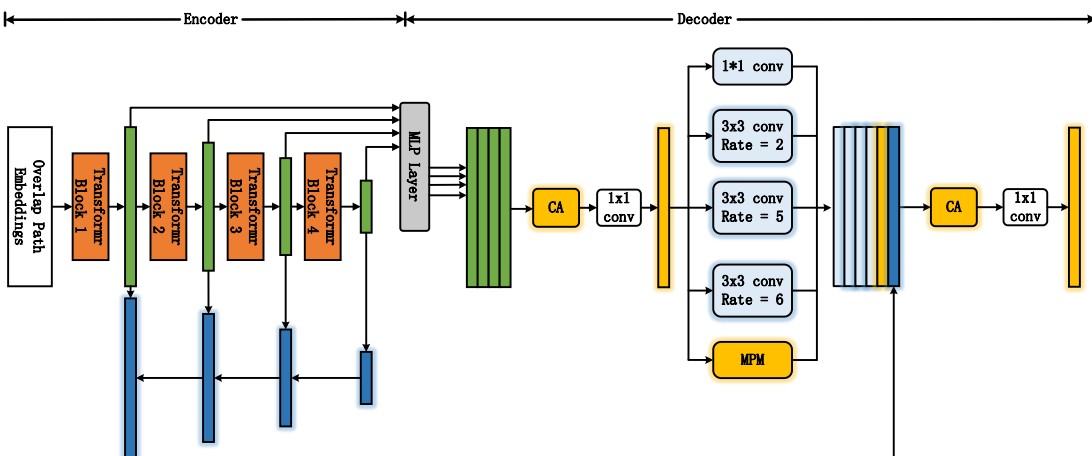

**Figure 6: The architecture of modified Segformer.**

For the improved ASPP module, we set the dilation rates of the atrous convolutions in the ASPP module to [2, 5, 6] to mitigate the grid effect (Wang et al., 2018) and accommodate oceanic and atmospheric phenomena of different scales. Additionally, the global average pooling layer in the ASPP has certain limitations, as it cannot fully capture the feature information of oceanic and atmospheric phenomena with varying shapes. Therefore, in this study, we replace the global average pooling layer in the ASPP module with a Mixed Pooling Model (MPM) (Hou et al., 2020), which combines different pooling methods to effectively capture both short-range and long-range dependencies in the feature maps. For the upsampling module, we employ a progressive upsampling method similar to that used in U-Net to fuse the four different scale feature maps extracted by Segformer. This approach allows the network to better utilize contextual information and reduces the information loss that occurs with direct upsampling in the original Segformer network. Additionally, we incorporate the CA mechanism module to enhance the fusion of feature maps, improving the segmentation capability for target regions.

### 3.2 Training Strategy

All experiments were carried out on NVIDIA GeForce RTX 3090 GPU by using the PyTorch framework. Regarding hyperparameter settings, the batch size for all experimental models is set to 16, and the models are trained for 80,000 iterations. The AdamW optimizer with a momentum of 0.9 is used to train the network, with an initial learning rate of 0.00006 and a weight decay set to 0.01.

Due to the differences in pixel counts for each phenomenon, we used a multi-loss function strategy to train the deep learning model. We combined weighted cross-entropy loss, Dice loss, and Focal loss to mitigate the impact of pixel value imbalance. We used the Dice coefficient (Dice) and Intersection over Union (IoU) as metrics to evaluate the positional differences between each phenomenon and the ground truth labels. Overall Accuracy (OA) represents the proportion of correctly classified pixels in the images.

Because oceanic and atmospheric phenomena influenced by different environmental factors have varying characteristics, a large and diverse dataset is crucial for achieving good segmentation results. Data augmentation methods can effectively expand our dataset, enhancing its diversity and improving the network's segmentation capabilities. Therefore, in this study, we augment the original dataset with several techniques, including horizontal and vertical flipping, image rotation, and photometric distortion.





**4 Result and Validation**
**4.1 Overall Evaluation Results**
To compare the segmentation performance of different models, we select four classic semantic segmentation models for
comparative experiments. To ensure the fairness of the experiments, all networks are trained in the same hardware environment
and obtain segmentation results based on the same training set, validation set, and test set. Table 1 presents the segmentation
results of each model on the test set.

**Table 1: Comparison of segmentation results of five models.**

|  | mDice (%) | mIoU (%) | OA (%) |
|---|---|---|---|
| **U-Net** (Ronneberger et al., 2017) | 72.31 | 59.29 | 79.07 |
| **DeepLabV3+** (Chen et al., 2018) | 78.81 | 68.04 | 84.93 |
| **SETR** (Zheng et al., 2021) | 78.21 | 67.50 | 84.81 |
| **Segformer** (Xie et al., 2021) | 78.83 | 68.08 | 85.20 |
| **Modified Segformer** (Ours) | **80.98** | **70.32** | **86.77** |


The modified Segformer model demonstrates the best performance, with scores of 80.98% (Dice), 70.32% (IoU), and 86.77%
(Accuracy). Compared to U-Net, the proposed model improves the average Dice score by 8.67%, IoU by 11.03%, and accuracy
by 7.76%. Compared to the baseline Segformer, the proposed model enhances the average Dice score by 2.15%, IoU by 2.24%,
and accuracy by 1.57%.
The visual inspection of the segmentation results is given in Figure 7. It confirms that the most promising method is the
modified Segformer. These observations are coherent with the segmentation results values.

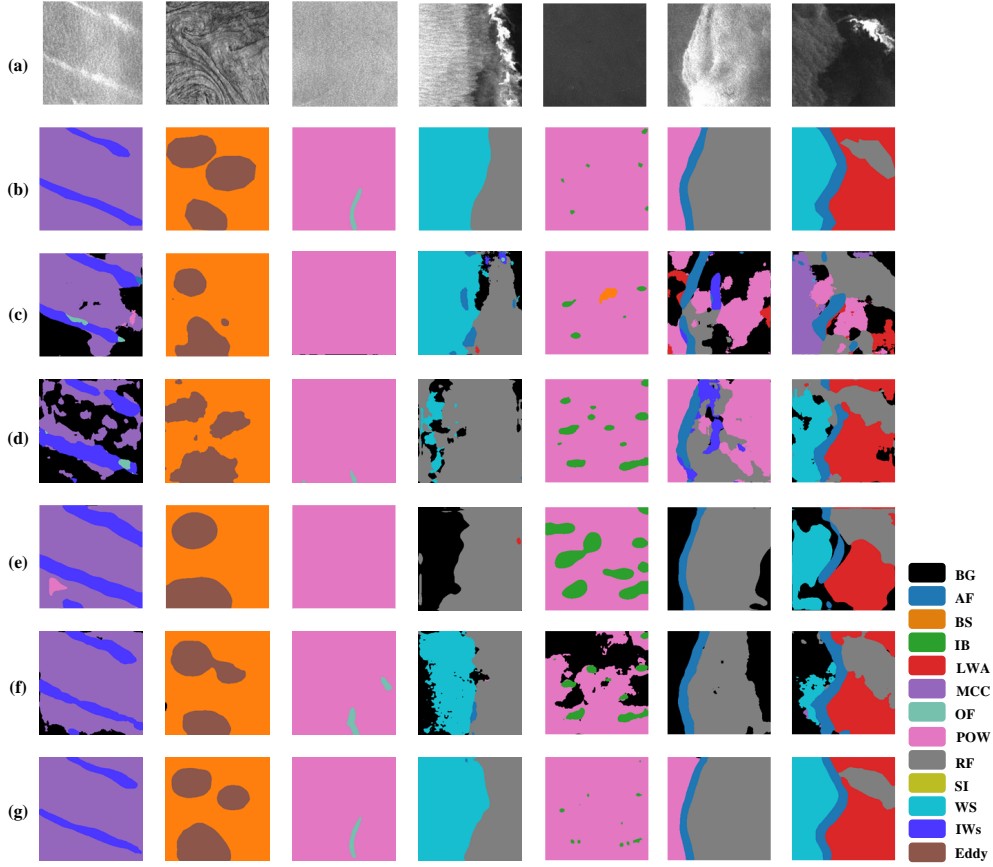

**Figure 7: Visualization of segmentation results of five models (a) SAR Images; (b) Ground truth; (c) U-Net; (d) DeepLabV3+; (e) SETR; (f) Segformer; (g) Modified Segformer.**

The results indicate that when multiple phenomena coexist in SAR images, U-Net and DeepLabV3+ frequently exhibit numerous misclassifications, unclear boundary segmentation, and severe image distortion. Additionally, due to the limitations of the receptive field of CNNs, these models cannot accurately identify large-scale, long-distance phenomena, resulting in lower segmentation accuracy. They also fail to delineate the contours of small-scale phenomena such as icebergs and ocean fronts. In contrast, the Transformer-based models SETR and Segformer show some improvement over the aforementioned models, achieving relatively accurate boundary recognition for various phenomena. However, they still encounter segmentation errors for small-scale and complex-featured phenomena. Notably, the modified Segformer model proposed in this study demonstrates superior segmentation capabilities for oceanic and atmospheric phenomena. It accurately segments the boundaries of different phenomena in complex scenes, improves the segmentation of small-scale phenomena with clear contours, and has a lower false alarm rate, producing results that closely align with the ground truth labels.

## 4.2 Segmentation results for different phenomena

Table 2 presents the Dice coefficients for the segmentation results of twelve oceanic and atmospheric phenomena across five networks, with the best segmentation results for each phenomenon highlighted in bold. The results show that the modified Segformer model proposed in this study achieves the best segmentation results for eight phenomena: ocean fronts, rainfall, icebergs, sea ice, pure ocean waves, wind streaks, oceanic internal waves, and Eddies. Compared to the baseline Segformer network, the detection of ocean fronts, icebergs, and eddies shows significant improvement, with Dice coefficients increasing





by 3.4%, 9.53%, and 6.28%, respectively, demonstrating the enhanced capability of the modified Segformer network in
extracting small targets and learning complex features. Although atmospheric fronts, biological slicks, low wind speed areas,
and micro-convective cells did not achieve the best segmentation results, the differences from the best results are minimal, at
0.07%, 0.17%, 0.06%, and 0.04% respectively, which are within an acceptable range. Overall, the modified Segformer model
exhibits the best comprehensive segmentation performance.

**Table 2: Dice coefficients (%) of segmentation of different phenomena by five models, the best results are shown in black bold.**

|  | AF | OF | RF | IC | SI | POW |
|---|---|---|---|---|---|---|
| U-Net | 49.35 | 57.06 | 74.84 | 46.1 | 95.02 | 79.9 |
| DeepLabV3+ | 61.17 | 64.81 | 84.17 | 44.01 | 99.31 | 84.33 |
| SETR | **63.18** | 63.86 | 87.17 | 34.93 | 99.27 | 83.05 |
| Segformer | 61.73 | 63.62 | 87.58 | 39.46 | 99.85 | 85.65 |
| Ours | 63.11 | **67.02** | **87.79** | **48.99** | **99.87** | **86.47** |
|  | WS | LWA | BS | MCC | IWs | Eddy |
| U-Net | 86.65 | 85.61 | 85.32 | 80.25 | 82.42 | 45.17 |
| DeepLabV3+ | 92.69 | 89.11 | 90.67 | 84.68 | 86.99 | 63.87 |
| SETR | 91.72 | **89.99** | **90.78** | **84.71** | 84.64 | 65.18 |
| Segformer | 91.44 | 89.54 | 90.60 | 84.5 | 86.1 | 65.92 |
| Ours | **94.08** | 89.9 | 90.61 | 84.67 | **87.08** | **72.2** |


Among the twelve typical oceanic and atmospheric phenomena, large-scale ocean phenomena such as rainfall, sea ice, pure
ocean waves, wind streaks, low wind speed areas, honeycomb convection, oceanic internal waves, and biological slicks have
relatively distinct features, resulting in better segmentation outcomes with Dice coefficients exceeding 80% for all five
networks. For phenomena with complex features, such as atmospheric fronts and ocean eddies, the models exhibit some
misclassification. Atmospheric fronts have multiple distinct features (Catto et al., 2014), making it difficult for the modified
Segformer model to learn all characteristics from a limited dataset. Ocean eddies present various forms due to different
formation mechanisms, such as "black eddies" and "white eddies" (Stuhlmacher and Gade, 2020), adding to the segmentation
challenge due to their feature diversity. Iceberg segmentation results are the lowest among the five networks, primarily because
icebergs are small in scale, often occupying only a few pixels in the image, which increases the difficulty for model
segmentation. Furthermore, Sentinel-1 IW mode images are collected near the coast, where human-made structures like ships,
which share similar characteristics with icebergs, are present, leading to a higher misclassification.

Notably, each network demonstrates excellent recognition capability for sea ice, with Dice coefficients exceeding 95%. This
high accuracy is primarily due to the large coverage area of SAR images and the distinct characteristics of sea ice. Additionally,
this study focuses on the basic features of sea ice without classifying its types, making the image segmentation task similar to
an image classification task.
**4.3 Comparison with visual interpretation results**
To validate the segmentation performance of the modified Segformer model on full Sentinel-1 IW mode images, we select
Setinel-1 IW mode image containing multiple phenomena for testing. First, the original image undergoes preprocessing steps.
The preprocessing results are shown in Figure 4-4. It is clear that the SAR image primarily contains three oceanic and





atmospheric phenomena: low wind speed areas, biological slicks, and micro-convective cells. Additionally, small-scale ocean
eddies are present, as indicated by the orange boxed sub-image area in Figure 8. The entire SAR image is then divided into
256 × 256 sub-images with a certain overlap rate and input into the modified Segformer model for testing. The segmentation
results are shown in Figure 9. The results clearly display the three primary oceanic and atmospheric phenomena in the SAR
image: the red mask represents low wind speed areas, the yellow mask represents biological slicks, and the purple mask
represents micro-convective cells. The modified Segformer model also successfully identifies and segments smaller-scale
ocean eddies, as indicated by the brown mask areas, achieving accurate segmentation results.

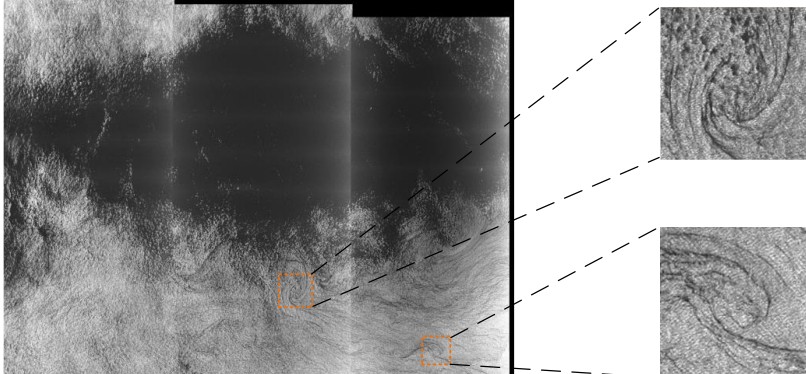

**Figure 8: Preprocessed Sentinel-1 IW mode image.**

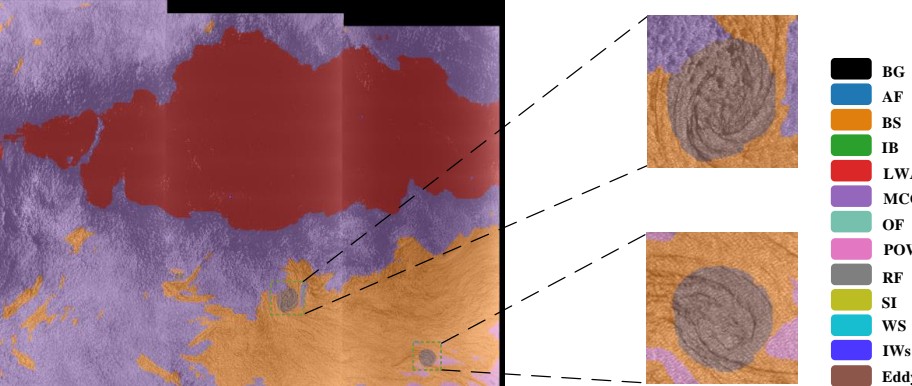

**Figure 9: Segmentation result display. The segmentation results are displayed overlaid with the original image.**

To validate the segmentation performance of the modified Segformer model on Sentinel-1 WV mode images, we select several
WV mode images containing typical oceanic and atmospheric phenomena for testing. As shown in Figure 10, the segmentation
results demonstrate that for large-scale phenomena such as pure ocean waves, low wind speed areas, biological slicks, and sea
ice, the proposed model accurately identifies and segments specific regions. Additionally, the model also performs well in
segmenting smaller-scale oceanic and atmospheric phenomena within the images, such as icebergs, ocean fronts, and ocean
eddies.

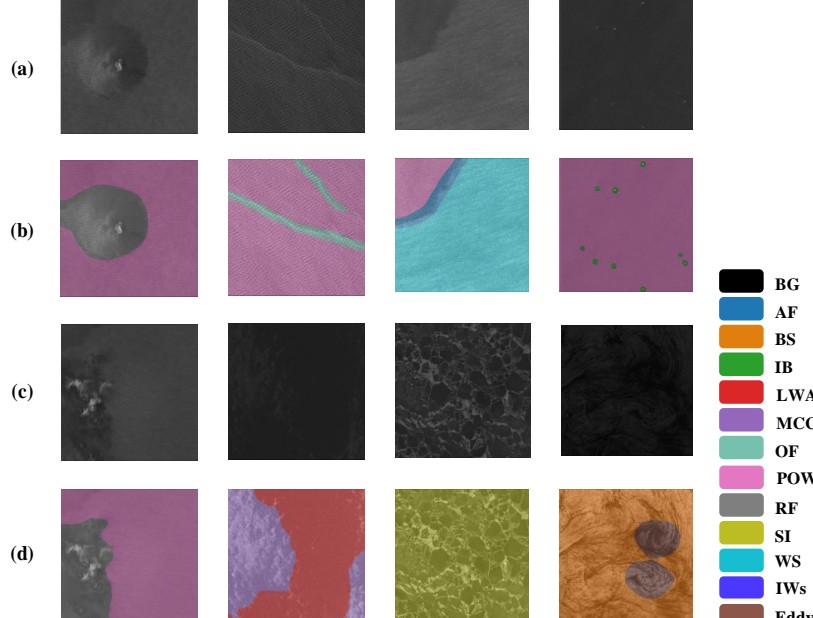

Figure 10: Sentinel-1 WV mode image segmentation Results. The segmentation results are displayed overlaid with the original image.

## 4.4 Case Study

### 4.4.1 Oceanic internal wave

We use the oceanic internal wave object detection dataset (Tao, 2022b) to validate the segmentation performance of the modified Segformer model on oceanic internal waves. Since our dataset construction references part of the dataset from Tao et al., to avoid data overlap, an additional set of images was selected. These images were captured on December 7, 2017, and April 9, 2020, in the Celebes Sea region. As shown in Figure 11, the green boxed areas in the images indicate the object detection annotations for oceanic internal waves.

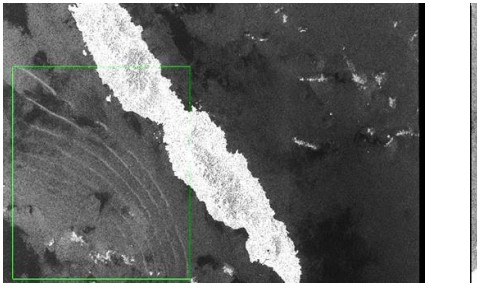
(a)
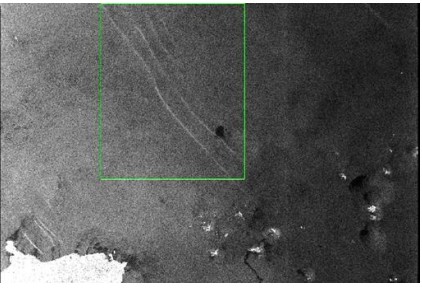
(b)

Figure 11: Internal Wave object detection data example (a) Case1; (b)Case 2.

First, the original Sentinel-1 IW mode images were downloaded and reprocessed. The preprocessing results are shown in Figure 12. Next, the entire IW mode image was cropped into sub-images with a certain overlap rate and input into the modified Segformer model for testing to obtain the final segmentation results. Figure 13 shows the overlay of the internal wave



segmentation results with the original images, where the colored areas represent the internal wave segmentation results.

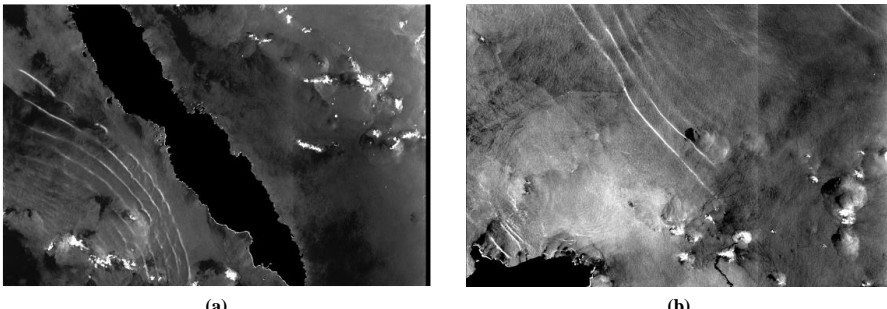

**Figure 12: Re-preprocessing result image (a) Case1; (b)Case 2.**

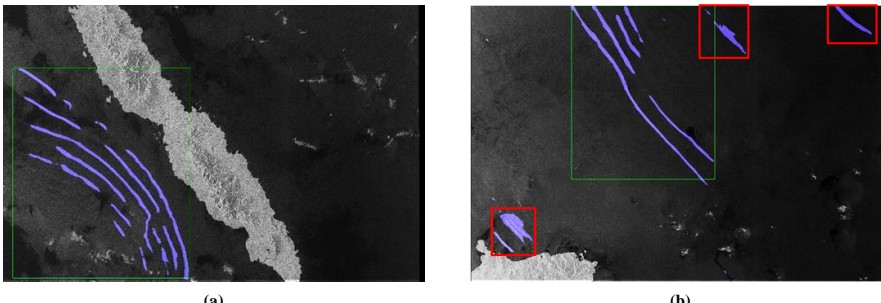

281                              (a)                                                    (b)
**Figure 13: Segmentation result display. The segmentation results are displayed overlaid with the original image (a) Case1; (b)Case**
**2.**
Comparing the object detection annotations in Figure 11 with the segmentation results in Figure 13, the modified Segformer
model can clearly and accurately extract internal wave stripes from complex oceanic and atmospheric phenomena, with the
extraction results consistent with the object detection labels. Additionally, the segmentation results reveal extra internal wave
stripes, as indicated by the red boxes in the figure. This demonstrates that the modified Segformer model not only can clearly
segment large-scale internal wave stripes but also performs well in segmenting individual internal wave stripes.
**4.4.2 Rainfall**
In this section, the modified Segformer model's segmentation results for rainfall phenomena are validated using IMERG data
(Pradhan et al., 2022), a Level 3 product from the GPM satellite. IMERG integrates and interpolates microwave precipitation
estimates, infrared precipitation estimates, and ground truth data to produce precipitation products with a temporal resolution
of 0.5 hours and a spatial resolution of 0.1°. This study extracts rainfall data from IMERG to characterize rainfall areas and
compare them with the model's segmentation results.
We select two typical IW mode image cases of Sentinel-1 containing rainfall phenomena, which were taken in the
Mediterranean area and the sea near Singapore on October 29, 2022 and October 30, 2022 respectively. The preprocessing
results are shown in Figure 14. As can be clearly seen from the figure, both selected SAR images exhibit noticeable rainfall
phenomena. After preprocessing, the images were cropped into sub-images and input into the modified Segformer model to
obtain the segmentation results for the rainfall phenomena. The overlay of the segmentation results with the original images is
shown in Figure 15, where the white areas represent the model's segmentation results. From the perspective of visual
interpretation, the modified Segformer model accurately segmented the rainfall phenomena over the ocean surface.



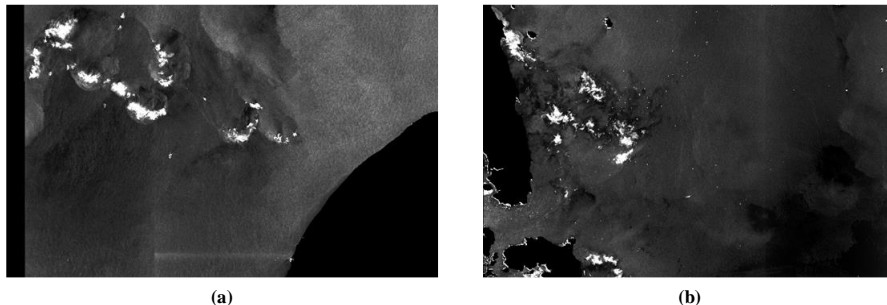

<p style="text-align:center">(a)                (b)</p>

**Figure 14: Preprocessed result image (a) Case1; (b)Case 2.**

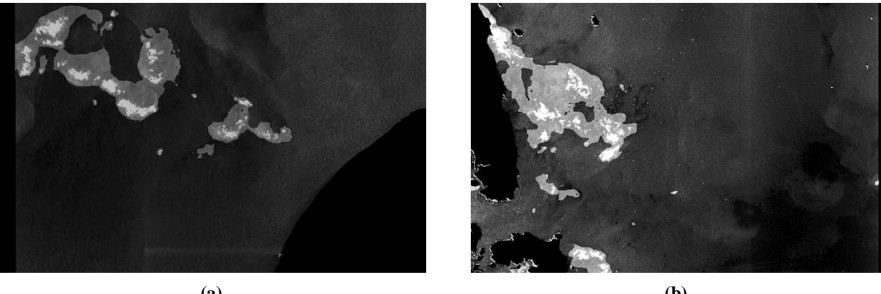

<p style="text-align:center">(a)                (b)</p>

**Figure 15: Segmentation result display. The segmentation results are displayed overlaid with the original image (a) Case1; (b)Case 2.**

The comparison between the model segmentation results and the GPM rainfall data is shown in Figure 16 and Figure 17. The rainfall areas identified by the GPM data closely correspond to the rainfall areas in the segmentation results. The slight differences observed may be due to the GPM IMERG product providing average rainfall data over a 0.5-hour period, during which the rainfall areas can change over time. In conclusion, the modified Segformer model proposed in this study can accurately identify rainfall areas over the ocean, demonstrating robust segmentation performance.

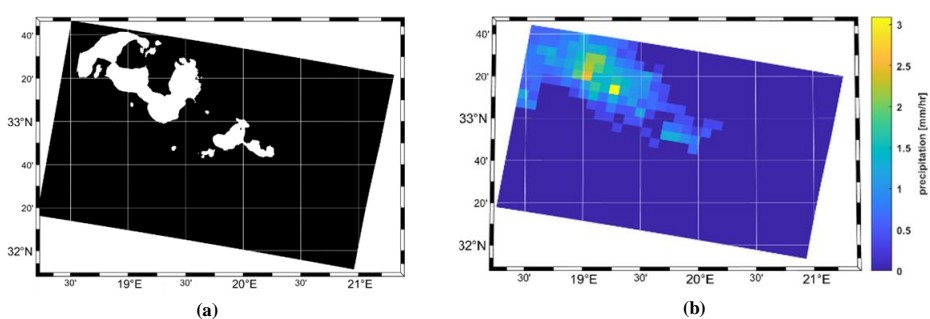

<p style="text-align:center">(a)                (b)</p>

**Figure 16: Comparison of the segmentation result image of Case 1 and GPM data (a) Segmentation result; (b) GPM data.**



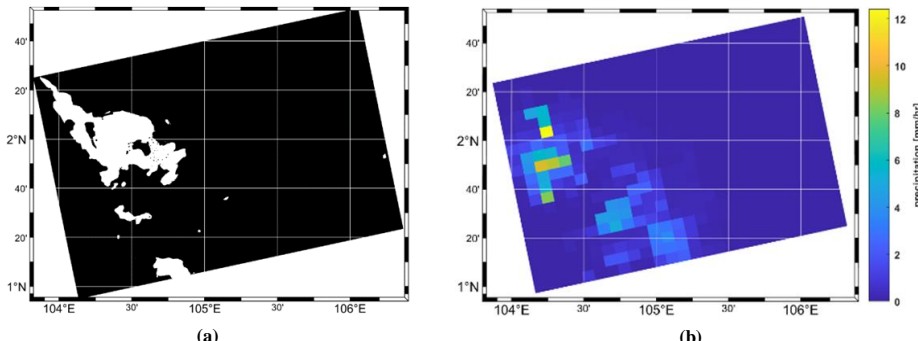

|  (a)  |  (b)  |

**Figure 17: Comparison of the segmentation result image of Case 2 and GPM data (a) Segmentation result; (b) GPM data.**

## 5 Data Availability

The dataset constructed in this paper can be downloaded from: https://doi.org/10.5281/zenodo.11410662 (Quankun et al., 2024). This dataset contains SAR images, Json files and PNG annotations.

Other public data used in this paper can be downloaded from the following website:

Sentinel-1: https://search.asf.alaska.edu/.

TenGeoP-SARwv: https://doi.org/10.17882/56796 (Wang et al., 2019a).

SAR_WV_SemanticSegmentation: https://www.kaggle.com/datasets/rignak/sar-wv-semanticsegmentation.

Internal Wave Dataset: https://doi.org/10.6084/m9.figshare.21365835.v3 (Tao et al., 2022b).

## 6 Summary and Conclusion

In this study, we used Sentinel-1 IW and WV mode data to construct a SAR image semantic segmentation dataset. This dataset includes twelve typical oceanic and atmospheric phenomena: atmospheric fronts, oceanic fronts, rainfall, icebergs, sea ice, pure sea waves, wind streaks, low wind speed areas, biogenic slicks, microwave convection cells, oceanic internal waves, and ocean eddies. Advanced deep learning algorithms have been employed to achieve the recognition and localization of these phenomena in SAR images.

The generated dataset of various oceanic and atmospheric phenomena could enhance our understanding of surface phenomena. It also provides valuable information for studying the interactions between multi-scale oceanic and atmospheric processes, promoting further research into ocean dynamics. Moreover, its high-resolution characteristics can supplement traditional observational data of atmospheric phenomena, aiding in the improvement of meteorological forecast models and enhancing their accuracy, which is vital for the development of high-resolution models. Furthermore, the availability of this extensive IW dataset is critical to advancing AI oceanographic research. It is the most comprehensive SAR image semantic segmentation dataset, covering the widest variety of oceanic and atmospheric phenomena. This dataset allows researchers to evaluate the segmentation performance of different models.

This paper focuses on the semantic segmentation task, which aims to classify each pixel in an image to segment different phenomena. This method assumes that each pixel belongs to a mutually exclusive category. However, in practice, different categories often overlap, meaning a single pixel may belong to multiple categories, increasing the complexity of segmentation. Introducing new categories that encompass mixed oceanic and atmospheric phenomena can help alleviate this issue to some extent. Additionally, the dataset consists of images with a resolution of 100m and dimensions of 256×256 pixels, covering relatively small areas (25km×25km for IW mode sub-images and 20km×20km for WV mode sub-images), which limits

segmentation accuracy for large-scale oceanic phenomena. Balancing training image size and model performance is crucial

for addressing this challenge. Employing multi-scale segmentation methods can aid in segmenting larger-scale phenomena.

Overall, the SAR image dataset proposed in this study makes a significant contribution to oceanography, providing valuable

data resources for studying the dynamic processes of multi-scale oceanic and atmospheric phenomena, validating deep learning

models, and developing high-resolution models. This dataset is anticipated to stimulate further research and advancements in

understanding the complex dynamics of sea surface.

**Competing Interests**

The contact author has declared that none of the authors has any competing interests.

**Author Contributions**

Conceptualization, Q.L.; writing—original draft preparation, Q.L. writing—review and editing, X.B., L.H., X.G. and X.-H.Y.;

visualization, Q.L. All authors have read and agreed to the published version of the manuscript.

**Acknowledgments**

This work was supported in part by the Industry-University Cooperation and Collaborative Ed-ucation Projects

(202102245034). Xiao-Hai Yan have been supported by NSF (IIS-2123264) and NASA (80NSSC20M0220). The authors

would like to thank Spacety, China Electronic Technology Group Corporation (CETC) 38th Institute and Fujian Jingwei Digital

Technology Co, Ltd for developing HISEA-1 satellite.

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
