# Peer review of "SAR Image Semantic Segmentation of Typical Oceanic and Atmospheric Phenomena"

_Earth System Science Data, 2024_

## Author Comment (AC1)

**Author response to reviewer comments**

**Anonymous Referee #1**

I appreciate the effort and work put into this manuscript, which focuses on constructing a dataset of SAR images annotated for 12 types of oceanic and atmospheric phenomena and developing a deep learning model to segment these phenomena. The paper addresses a significant topic and provides valuable contributions. However, several areas need to be addressed to improve the overall quality and clarity of the manuscript.

Thank you for your valuable comments and suggestions on our manuscript. Your detailed feedback has been instrumental in helping us improve the quality of our work. Below, we provide a detailed response to each of your specific comments and the corresponding revisions we have made.

Review comments in blue

Reply in black

**1 Dataset:**

Q1. The criteria for determining the boundaries of each phenomenon are not clearly defined. For example, the internal wave is identified by its wave crest lines, while the pure ocean wave includes both wave crest lines and surrounding seawater. The boundary size for eddies is not clearly defined, and typically, eddies detected by SAR are accompanied by biological slicks, which are not considered in the dataset.

This study focuses on 12 typical oceanic and atmospheric phenomena. For the 10 oceanic and atmospheric phenomena with existing research, we referenced the segmentation standards of Benchaabane et al., Wang et al. and Colin et al. (Benchaabane et al., 2022; Colin et al., 2022; Wang et al., 2019a). For the two newly added oceanic phenomena, oceanic eddies and oceanic internal waves, we established segmentation standards based on relevant literature, as follows:

1)  For oceanic eddies: Oceanic eddies change sea surface roughness by carrying tracers (such as

sea ice and biological slicks) or affecting surface flow fields, creating distinct elliptical patches or bands on SAR images. Depending on their formation mechanisms, they are primarily categorized as "dark eddies" and "white eddies"(Ji et al., 2021; Kozlov et al., 2019; Stuhlmacher and Gade, 2020). In this manuscript, the minimum enclosing shape of the eddies is used as the ground truth label. Notably, biological slicks often serve as tracers for oceanic eddies, and in overlapping cases, the priority of identifying the eddy phenomenon is higher than that of the biological slick phenomenon.

2)  For oceanic internal wave phenomenon: Oceanic internal waves appear in SAR images as irregular stripes of alternating light and dark patterns. To ensure accurate labeling, we referred to publicly available object detection datasets. (Tao et al., 2022a)for the annotation.

Q2. The sea ice regions in the images seem to include ice leads, yet the entire area is labeled as sea ice. Additionally, the separation between low wind speed areas and biological slicks or oil spills is not clearly explained.

Regarding the sea ice you mentioned, the marking of sea ice in the manuscript follows a similar logic to that of Wang et al.(Wang et al., 2019b), the goal is to distinguish sea ice from open water in the SAR images. Thus, the segmentation label creation focuses on delineating areas and does not achieve the precision required to segment out sea leads. We will continue to improve and refine the sea ice labeling.

For distinguishing low wind speed areas, biological slicks, and oil spills: Low wind speed areas often coexist with biological slicks. Low wind speed regions are characterized by large dark patches on the sea surface, while biological slicks appear as aggregated black filaments (Najoui et al., 2018). Oil spills typically manifest as isolated black filaments, and relevant oil spill datasets will be provided later.

Q3. Internal waves and eddies, particularly eddies, typically occur offshore. Using IW mode data limits the representation of these phenomena.

For oceanic eddy phenomena, to ensure data diversity, we selected images from both the Sentinel-1 IW mode and the WV mode from the TenGeoP-SARwv dataset, with a roughly equal number of images from each mode. (Although the TenGeoP-SARwv dataset does not include a category for oceanic Eddy, its classification of biological slicks contains a substantial number of oceanic Eddy phenomena.)

Regarding the oceanic internal wave images in the WV mode, their size limits the assessment of these waves (Colin et al., 2022). To ensure the accuracy of the evaluation, we use the Sentinel-1 IW mode ocean internal wave object detection dataset proposed by Tao et al. (Tao et al., 2022b) for data labeling.

Q4. Additionally, the manuscript mentions using 484 IW images to select samples of internal waves and eddies, but it is unclear where these images are located, how representative they are, and the criteria for their selection.

The IW mode images are divided into two parts. One part comes from the ocean internal wave dataset, while the other part consists of images from Sentinel-1 IW mode, selected from ASF (https://search.asf.alaska.edu/), featuring typical oceanic and atmospheric phenomena.

For the oceanic internal wave phenomenon, we selected images consistent with the dataset, which includes Andaman Sea, South China Sea, Sulu Sea, and Celebes Sea area (Tao et al., 2022b). We randomly selected 50 IW mode images from each study area for annotation.

For other phenomena, we selected Sentinel-1 IW mode images from 2021-2022 that exhibit typical characteristics of oceanic and atmospheric phenomena for annotation. We have added a distribution map of these SAR images in the revised manuscript.

[Figure]

**Figure: The data distribution (red is WV mode, blue is IW mode) and the number of images for each category.**

Q5. The TenGeoP-SARwv dataset (Wang et al., 2019), on which this study builds, does not provide geographical information, making it difficult for readers or users to assess the representativeness of the images.

In this study, we randomly selected images from the TenGeoP-SARwv dataset according to different

oceanic and atmospheric phenomena when constructing the dataset. We have updated the dataset to ensure it includes geographical information.

Q6. Except for the final rainfall image, the manuscript does not provide geographical coordinates for all the SAR images.

We have modified the SAR images in the manuscript to ensure that each image fully displays the latitude and longitude coordinate information.

**2 Metric Calculation:**

Q1. Many of the selected phenomena, such as fronts, internal waves, and icebergs, are significantly smaller in pixel count compared to the background (seawater). The manuscript does not exclude the background when calculating metrics, leading to potentially inflated performance scores.

In this manuscript, we have already excluded the background category when calculating the metrics. Taking mDice as an example, we first calculate the Dice coefficient for each category, and then we average the results for the 12 phenomena of interest, thus obtaining the mDice for these 12 phenomena, excluding the background category.

Q2. However, in the case of internal wave extraction (Figures 12 and 13), several rain cells are visible but not identified by the model.

In this manuscript, we selected ocean internal waves and rainfall phenomena for visual validation. To avoid interference from other phenomena, Figures 12 and 13 show only the segmentation results for ocean internal waves. The complete images of the following results are provided. The figures demonstrate that the rainfall phenomenon was successfully detected and identified, but it interfered with the visual interpretation of the ocean internal waves. Therefore, we have retained only the ocean internal wave phenomenon for analysis.

[Figure]

[Figure]

| | |
|---|---|
| ■ | BG |
| ■ | AF |
| ■ | BS |
| ■ | IB |
| ■ | LWA |
| ■ | MCC |
| ■ | OF |
| ■ | POW |
| ■ | RF |
| ■ | SI |
| ■ | WS |
| ■ | IWs |
| ■ | Eddy |

**Figure: Complete test results (Figures 12 and 13). (The images are consistent with those in the first draft of the manuscript to facilitate comparison.)**

Q3. The manuscript should include a comparison with ground truth and corresponding metrics for all phenomena.

In this manuscript, Figure 8 compares the segmentation results of five networks on the test set with the ground truth, while the corresponding metrics for all phenomena are presented in Table 2.

Q4. Additionally, the rationale for selecting only internal waves and rain cells for demonstration should be clarified (Section 4.4).

In SAR images, the overlap of different phenomena often occurs, which poses significant challenges

for semantic segmentation tasks. We selected oceanic internal waves and rainfall—two typical oceanic and atmospheric phenomena—for validation because they cause substantial changes in sea surface roughness, making overlap with other phenomena less likely. This ensures a clearer visual interpretation and more reliable assessment of the segmentation results. We have already stated this in the article.

Q5. The use of GPM half-hour rainfall data introduces temporal and spatial discrepancies with SAR imaging, which should be acknowledged. Figure 16 illustrates a noticeable discrepancy in the center location of the rainfall. It is recommended to define the criteria for identifying rain cells and directly compare them with ground truth rather than relying on GPM data.

We have restated in the article that there are certain discrepancies between the GPM rainfall data and SAR images. Regarding your mention of comparing dynamic annotations with segmentation results, we have performed similar comparison validation on the test set (Figure 8). The main objective of this section is to perform comparative validation with external data. Comparing manually labeled annotations with segmentation results may introduce a certain level of subjectivity, making it challenging to ensure the accuracy of the segmentation results.

**3 Others**

Q1. Geographical Coordinates and Imaging Time: Remote sensing images should include geographical coordinates and imaging time, which are crucial in geoscience research.

We have updated the SAR images in the manuscript to accurately display the geographic coordinates and the time of image capture.

Q2. Terminology and Labeling: On line 289, page, 'individual' might not be accurate. It is a group approaching the shore (Figure 13b).

We have revised the inaccurate statements accordingly. (L329)

Q3. The abbreviation IW is ambiguous and can refer to both Sentinel-1 imaging mode and internal waves.

We have made revisions to the manuscript. "IW" refers to the Sentinel-1 IW mode, while "IWs" stands for Internal Waves.

Q4. The term BG in the figures is not explained in the text.

"BG" refers to the background classification, which indicates the sea surface outside the phenomena of interest. We have included a definition of this in the manuscript.

Q5. Units and numbers should have a space in between (e.g., lines 348 and 349, page 14).

We have made the modifications in the manuscript. (L387)

**Reference**

Benchaabane, A., Peureux, C., and Soulat, F.: A labelled dataset description for SAR images segmentation, 2022.

Colin, A., Fablet, R., Tandeo, P., Husson, R., Peureux, C., Longépé, N., and Mouche, A.: Semantic Segmentation of Metoceanic Processes Using SAR Observations and Deep Learning, Remote Sensing, 14, 851, https://doi.org/10.3390/rs14040851, 2022.

Ji, Y., Xu, G., Dong, C., Yang, J., and Xia, C.: Submesoscale eddies in the East China Sea detected from SAR images, Acta Oceanol. Sin., 40, 18–26, https://doi.org/10.1007/s13131-021-1714-5, 2021.

Kozlov, I. E., Artamonova, A. V., Manucharyan, G. E., and Kubryakov, A. A.: Eddies in the Western Arctic Ocean From Spaceborne SAR Observations Over Open Ocean and Marginal Ice Zones, J. Geophys. Res. Oceans, 124, 6601–6616, https://doi.org/10.1029/2019JC015113, 2019.

Najoui, Z., Riazanoff, S., Deffontaines, B., and Xavier, J.-P.: A statistical approach to preprocess and enhance c-band SAR images in order to detect automatically marine oil slicks, IEEE Trans. Geosci. Remote Sens., 56, 2554–2564, 2018.

Stuhlmacher, A. and Gade, M.: Statistical analyses of eddies in the Western Mediterranean Sea based on Synthetic Aperture Radar imagery, Remote Sensing of Environment, 250, 112023, 2020.

Tao, M., Xu, C., Guo, L., Wang, X., and Xu, Y.: An Internal Waves Data Set From Sentinel-1 Synthetic Aperture Radar Imagery and Preliminary Detection, Earth Space Sci., 9, e2022EA002528, https://doi.org/10.1029/2022EA002528, 2022a.

Tao, M., Xu, C., Guo, L., Wang, X., and Xu, Y.: An Internal Waves Data Set From Sentinel-1 Synthetic Aperture Radar Imagery and Preliminary Detection, Earth Space Sci., 9, e2022EA002528, https://doi.org/10.1029/2022EA002528, 2022b.

Wang, C., Mouche, A., Tandeo, P., Stopa, J. E., Longépé, N., Erhard, G., Foster, R. C., Vandemark, D., and Chapron, B.: A labelled ocean SAR imagery dataset of ten geophysical phenomena from Sentinel-1 wave mode, Geosci. Data J., 6, 105–115, https://doi.org/10.1002/gdj3.73, 2019a.

Wang, C., Tandeo, P., Mouche, A., Stopa, J. E., Gressani, V., Longepe, N., Vandemark, D., Foster, R. C., and Chapron, B.: Classification of the global Sentinel-1 SAR vignettes for ocean surface process studies, Remote Sens. Environ., 234, 111457, 2019b.

---

## Author Comment (AC2)

**Author response to reviewer comments**

**Anonymous Referee #2**

Thank you for your valuable suggestions regarding this manuscript. Your patience and attention to detail are truly impressive. The revision has been done based on your comments, and we believe that they have been addressed adequately and thoroughly in the revised manuscript. The point-by-point responses are shown below.

Review comments in blue

Reply in black

**1 Major comments:**

Q1. The manuscript focuses on the validation of a machine-learning model. This seems to be beyond the scope of this journal and I recommend the work be submitted elsewhere. The manuscript also creates a dataset - this is relevant to the journal; however, the dataset details are not sufficient to make it useful to others in the community. For example, how were the images selected? Are these images representative of the population? How many examples of each class are available? Is that number sufficient statistically (or enough for developing AI/ML models)? How was the labeling done (what were the guidelines)?

The main contribution of this manuscript is the semantic segmentation dataset for detecting and locating oceanic and atmospheric phenomena in SAR images. The primary goal of the machine learning model proposed in this study is to demonstrate the usability of the dataset and provide application suggestions for other researchers. In response to the issue of insufficient dataset description, we have revised the section on dataset construction to ensure a more comprehensive and detailed explanation.

Q1.1 How were the images selected?

We selected the labeled images based on the primary characteristics of each phenomenon, ensuring

no overlap and prioritizing representative samples. For WV-mode images, we selected 1,000 samples from Wang et al.'s TenGeoP-SARwv dataset (Wang et al., 2019) for annotation, and also included around 1,000 images annotated by Colin et al (Colin et al., 2022). Additionally, we selected IW-mode images from 2015 to 2022 and combined them with a subset of images from the internal wave target detection dataset for manual annotation.

**Q1.2 Are these images representative of the population?**

Our image selection was based on the definitions established in previous studies, which identified 10 major oceanic features (Benchaabane et al., 2022; Colin et al., 2022; Topouzelis and Kitsiou, 2015; Wang et al., 2019). For the two newly added oceanic phenomena, oceanic eddies and oceanic internal waves, we established segmentation standards based on relevant literature, as follows:

1) For oceanic eddies: Oceanic eddies change sea surface roughness by carrying tracers (such as sea ice and biological slicks) or affecting surface flow fields, creating distinct elliptical patches or bands on SAR images. Depending on their formation mechanisms, they are primarily categorized as "dark eddies" and "white eddies"(Ji et al., 2021; Kozlov et al., 2019; Stuhlmacher and Gade, 2020). In this manuscript, the minimum enclosing shape of the eddies is used as the ground truth label. Notably, biological slicks often serve as tracers for oceanic eddies, and in overlapping cases, the priority of identifying the eddy phenomenon is higher than that of the biological slick phenomenon.

2) For oceanic internal wave phenomenon: Oceanic internal waves appear in SAR images as irregular stripes of alternating light and dark patterns. To ensure accurate labeling, we referred to publicly available object detection datasets. (Tao et al., 2022)for the annotation.

**Q1.3 How many examples of each class are available?**

The table below presents the number of images for each phenomenon of interest discussed in this paper. Note that the total does not add up to 5,011, as a single image may contain multiple phenomena.

| Phenomenon | Total |
|---|---|
| AF | 644 |
| BS | 1034 |
| IB | 398 |
| LWA | 601 |
| MCC | 830 |
| OF | 437 |
| POW | 1746 |
| RF | 484 |
| SI | 454 |
| WS | 585 |
| Eddy | 501 |
| IWs | 417 |

**Figure: The number of images for each phenomenon.**

Q1.4 Is that number sufficient statistically (or enough for developing AI/ML models)?

The figure below shows the training loss curve and training accuracy. As shown in the figure, the curves stabilize, indicating that the dataset enables the model to effectively learn the features of the relevant phenomena. Additionally, based on the work of Colin et al., we expanded the dataset fourfold to ensure its comprehensiveness.

[Figure]

**Figure: Training loss curve and accuracy.**

Q1.5 How was the labeling done (what were the guidelines)?

In the manuscript, the cropped sub-images were annotated using the Labelme software based on the 8-bit SAR images. we referred to the definitions and annotation guidelines for these phenomena provided by previous studies (Benchaabane et al., 2022; Colin et al., 2022; Wang et al., 2019). We have modified the dataset construction section and provided references to the guidelines followed for annotation. (L134-Section 2.3)

Q2. A major limitation of the dataset and model is that multiple phenomena are not possible in one pixel. There are several statements in the manuscript suggesting that multi-tagging is possible... so, it is confusing.

In semantic segmentation tasks, each pixel is typically assigned a single label, meaning it can only belong to one category. In oceanic systems, some oceanic and atmospheric phenomena may overlap, but typically, there is a dominant phenomenon. The definitions of each phenomenon type in this manuscript are primarily based on the dominant phenomenon. (L382 - L383)

Q3. More references and clarifications are needed. The manuscript is difficult to follow because the statements are ambiguous. Several examples listed in the specific comments below identify statements that could be improved. Mainly the Introduction and Conclusion could use more references to provide the appropriate context.

Based on your suggestions, we have added additional references and optimized the paragraphs and language to avoid ambiguity, making the manuscript easier to understand.

Q4. All Figures and Tables should be referenced in the text.

We have added references and descriptions for all images and tables in the latest version of the manuscript.

Q5. The major finding is that the modified Segformer model performs better than the other models. However, it is unclear how hyperparameter tuning would influence the results.

We have adjusted several hyperparameters, including batch size, learning rate, and optimizer, to achieve optimal learning performance under the current hardware conditions. For the additional modules introduced in this paper, we conducted ablation experiments to determine the impact of the various modules added to the decoder section of the modified Segformer on the segmentation results. Below are the results of the ablation experiments:

**Table:Ablation Experiment Results (Results on the test set.).**

|  | mDice(%) | mIoU(%) | OA(%) |
|---|---|---|---|
| Exp1:Base Segformer | 78.83 | 68.08 | 85.20 |
| Exp 2:Base + ASPP | 79.46 | 68.59 | 85.39 |
| Exp 3:Base + ASPP + MPM | 79.92 | 69.18 | 85.45 |
| Exp 4:Base + ASPP + MPM + CA | 80.31 | 69.71 | 86.41 |
| Exp 5:Base + ASPP + MPM + CA + step-by-step upsampling | **80.98** | **70.32** | **86.77** |

**2 Specific comments:**

Q1. L12 - Are you referring to the ocean surface as observed by SAR? Otherwise, it is obvious that the ocean surface exhibits ocean phenomena.

Here, we are referring to the sea surface observed by SAR. This has already been clarified in the abstract. (L12)

Q2. Why is automatically detecting phenomena crucial? The statement is unjustified. Are you referring to the size of the SAR data?

Automatic detection of typical oceanic and atmospheric phenomena using SAR is essential due to the large volume and complexity of SAR data. Automation ensures efficient, accurate, and timely analysis, supporting applications like storm tracking, wave monitoring, and environmental assessments.

Q3. L18 - Maybe some readers are unfamiliar with the "average dice coefficient" so these 2 statements might not be the best way to communicate the results. I suggest generalizing your findings in the abstract better.

We have reorganized the abstract to more clearly convey the research findings of this paper. (L12-L20)

Q4. L22 - change matter to mass and add a reference or two.

OK! We have revised and added references. (L22)

Q5. L27 - why is remote sensing efficient? It covers space well but not time! It is expensive to develop, launch, maintain a satellite system and datasets.

Yes, the time revisit period of a single low Earth orbit satellite is not high, and the cost increases significantly when using multiple satellites. Here, "efficient" refers to the effectiveness of remote sensing methods in achieving extensive image coverage. We have updated the relevant descriptions to clarify this. (L26-L27)

Q6. L30-31: add or 2 a reference to support this statement.

Sure! We have added references. (L31)

Q7. L35-38: add a reference 2 to support this statement.

OK! No problem! (L39)

Q8. L44: numerous studies - but only 1 is listed. I suggest adding more references here.

We have added more references. (L44)

Q9. L47: good motivation - but what studies are you referring to? Add references

Thank you for your advice, we have added more references. (L47)

Q10. L59: "we think that using only 100 manually annotated samples for each phenomenon is insufficient to achieve the best segmentation results".

->This might be so… but in scientific journals, opinions should be supported by evidence or prior objective studies that support that statement. Please revise

We overlooked this issue, and we have updated the description accordingly. (L59-60)

Q11. L65 and L69-71 are inconsistent. The goals and the sections should be consistent.

We have revised the manuscript to ensure that the goals consistent with the sections.

Q12. L78 - Why are these phenomena "typical"? I expect that their occurrence in the open ocean is rare.

Many studies mention the application of SAR for observing ocean eddies and internal waves. Here, "typical" refers to ocean phenomena that SAR frequently observes.(Topouzelis and Kitsiou, 2015)

Q13. L82- wind streaks

Thank you very much for your careful review.

Q14. L95 - How were the 2383 WV images selected? How does this sample affect your results?

The 2,383 WV mode images include 1,100 images from the Colin dataset and a total of 1,283 images randomly selected from TenGeoP-SARwv for each phenomenon. The selection of WV mode images

was guided by relevant studies (Benchaabane et al., 2022; Wang et al., 2019), enhancing annotation accuracy and minimizing errors caused by misjudgment during labeling. The WV-mode images are representative of the world's open oceans.

[Figure]

**Figure: The data distribution (red is WV mode, blue is IW mode)**

Q15. L97 Figure 2 - the contrast is poor and it is difficult to view the phenomena. I question if the pre-processing described in Wang et al., (2019) (referenced in L103) was applied correctly.

Figure 2 shows the images we selected from TenGeoP-SARwv (16-bit). However, the 16-bit images used for training are not the same as the 8-bit images used for visual interpretation. We revised the manuscript to ensure the correct citation of the 8-bit images, make these images easier to follow

Q16. L101 - How were the 484 IW images selected? How does this sample affect your results?

Overall, the IW mode images are divided into two parts. One part comes from the ocean internal wave dataset, while the other part consists of images from Sentinel-1 IW mode, selected from ASF (https://search.asf.alaska.edu/), featuring typical oceanic and atmospheric phenomena.

For the oceanic internal wave phenomenon, we selected images consistent with the object dataset, which includes Andaman Sea, South China Sea, Sulu Sea, and Celebes Sea area (Tao et al., 2022). We randomly selected 50 IW mode images from each study area for annotation.

For other phenomena, we selected Sentinel-1 IW mode images from 2021-2022 that exhibit typical characteristics of oceanic and atmospheric phenomena for annotation. We have added a distribution map of these SAR images in the revised manuscript.

The inclusion of IW images complements the WV mode by providing coverage of coastal areas that WV cannot capture, thereby increasing data diversity. It also incorporates phenomena typically observed in IW mode, enabling a more comprehensive analysis of sea surface phenomena.

Q17. L123 - 8bit vs 16bit - Why is this distinction important?    How does the digit precision impact the results? This information is distracting if it is unimportant.

We adopted the preprocessing method from Wang et al. (Wang et al., 2019), using 8-bit images for visual interpretation to provide labels, while 16-bit images are used for model training. The 8-bit processing method enhances the contrast of the images, whereas the 16-bit processing method ensures that all texture and radiometric information is preserved in digital form.

Q18. L122 -256x256 subimages - Why? This window size (256*100 m) might not resolve all phenomena. This spatial scale limits the model development and output by not considering features larger than ~25 km.

The WV-mode typically has a coverage of around 20 km, to maintain data consistency, we followed the setup provided by Wang et al. (Wang et al., 2019)and Colin et al. (Colin et al., 2022). Additionally, hardware limitations of the equipment restricted the application of larger windows. Oceanic and atmospheric phenomena larger than 25 km are currently beyond the scope of this study. However, with the availability of more powerful hardware and wider bandwidth SAR images in the future, they may be addressable.

Q19. L131 Figure 5: Label all subpanels. Describe all subpanels in the text. BG - I guess that means background?

(a) This is likely not an atmospheric front but rather an atmospheric gravity wave.

(c) regions of slicks are also low wind areas

How would this multi-tagging influence the results?    This seems to be a potential issue of this approach.

(c third one from the left) The POW might be WS but the contrast makes it difficult to decipher.

We have labeled all subpanels and described them in the text. First, we apologize for not clearly describing the labels. "BG" represents the background. Second, the marked feature in the first image of row (a) is an ocean internal wave, identified based on the ocean internal wave target detection dataset. Next, in row (c), for the biogenic slick and low-wind-speed areas, we marked large black areas as low-wind-speed zones and thin black streaks as biogenic slicks, following the criteria in

(Benchaabane et al., 2022). Lastly, regarding the identification of POW and WS, we present an enlarged version of the image here, where POW is identified according to the specified criteria. (The wavelength of WS is longer than that of POW.)

[Figure]

**Figure: Figure 6 (c3).**

Q20. L136 "improved" relative to what? This statement seems to be comparing this approach to a previous approach. Please clarify or revise the statement.

Here, the comparison is with the original Segformer network. We have revised the manuscript to clarify it as "modified".

Q21. L155 "better utilize contextual information" - Maybe this is true but it is not common knowledge. I suggest adding references to support these claims of why the model setup is more appropriate.

OK! We have added some references to support this claim. (L179)

Q22. L172 the idea of augmentations is good. However, how the augmentations are implemented is unclear. My concern is that some of the augmentations might not be feasible depending on the physical environment and satellite trajectories.

In this manuscript, we applied data augmentation methods including random rotation and random flipping and photometric distortion. (L200-L202)

Q23. L182 Table 1 & L217 Table 2- how does the hyperparameter tuning influence the results? I am unsure if this is a fair comparison between models.

To ensure a fair comparison between the models, we used fixed hyperparameters for training all five networks. For example, the same batch size, learning rate, and optimizer were applied to all models. This approach ensures that performance differences between the networks are solely attributed to

the model architecture, rather than variations in hyperparameter settings, thereby enhancing the fairness of the comparison.

Q24. L191 Figure 7 - The poor contrast in the iceberg example might be making the problem very difficult for the models. Q25. L226 small icebergs - I expect the reason is the poor contrast in the images rather than the size of the target.

Low contrast also affects the model's performance. Initially, we used 16-bit images, which had low visual contrast and were difficult to interpret. After switching to 8-bit images, the contrast improved, but issues such as low contrast and small target size remain, making detection more challenging. We have added the relevant descriptions in the manuscript. (L257 - L259)

Q26. L241 What is the overlap rate?

To ensure that the image fully captures the phenomena to be identified, we processed the Sentinel-1 IW images with a stride of 32 during testing, allowing for maximum coverage of each image.

Q27. L240 Figure 8 - there are likely mixed rolls and cells in the bottom left of the image. It does not seem that the combination of these phenomena is possible in your approach. There might be more eddies than the 2 noted in your figure.

Here, we present a larger image, and from a visual interpretation perspective, no additional eddies were observed. However, there is significant overlap in the bottom-left corner of the image (MCC and BS), which increases the difficulty for the model in recognizing these features. Indeed, the model can only identify the dominant phenomenon.

[Figure]

Figure: Figure 9. (This image is consistent with the one in the initial draft of the manuscript to facilitate comparison.)

Q28. L254: How are these images selected? They look mostly homogenous with 1-2 phenomena each.

The WV mode images are from the test set. In real-world scenarios, most 256x256 images contain only a single category. For our testing, we selected multi-category images that encompass all detected phenomena, except for internal waves, which are exclusively observed in IW mode.

Q29. L259 Figure 10 - the image contrast is poor. This likely makes it more difficult for your models. Figure 13b - It seems the smaller-scale internal waves are not tagged (in the hand-tagged image) or found by the model.

First, the 16-bit images have very low contrast, so we switch to 8-bit images for display. Second, in Figure 13b, the green boxes indicate annotations from the object detection dataset (Tao et al., 2022), while the red boxes highlight additional detected internal waves (our method). We have enhanced the description of Figure 13b.

Q30. L306 Figure 15: Use a color of the detected regions that are not in the grey scale.    Figure 13b -It is difficult to interpret now. It seems that cold pools (old rain) are not labeled in the archive or captured by the model.

We have updated the annotation colors in Figure 15. In this manuscript, we selected ocean internal waves and rainfall phenomena for visual validation. To avoid interference from other phenomena, Figures 12 and 13 show only the segmentation results for ocean internal waves. The complete images of the following results are provided. The figures demonstrate that the rainfall phenomenon was successfully detected and identified, but it interfered with the visual interpretation of the ocean internal waves. Therefore, we have retained only the ocean internal wave phenomenon for analysis.

[Figure]

[Figure]

| | |
|---|---|
| ⬛ | BG |
| 🟦 | AF |
| 🟧 | BS |
| 🟩 | IB |
| 🟥 | LWA |
| 🟪 | MCC |
| 🟩 | OF |
| 🟪 | POW |
| ⬜ | RF |
| 🟨 | SI |
| 🟦 | WS |
| 🟦 | IWs |
| 🟫 | Eddy |

**Figure: Complete test results (Figures 12 and 13). (The images are consistent with those in the first draft of the manuscript to facilitate comparison.)**

Q31. L335-342: Ambitions are high but most statements are not justified or linked to the literature on the topic.

Thank you very much for your suggestion! We have revised this section to make the description more aligned with the main topic.

Q32. L346: It does not seem that multiple tags are possible for a given pixel. Please clarify

Got it! We have clarified this in the manuscript. (L382)

Q33. L353: What is the contribution to oceanography? The work is focused on validating a tool(/model) that can be applied to physical science problems.

We have rewritten Section 6 for clarity.

**Reference**

Benchaabane, A., Peureux, C., and Soulat, F.: A labelled dataset description for SAR images segmentation, 2022.

Colin, A., Fablet, R., Tandeo, P., Husson, R., Peureux, C., Longépé, N., and Mouche, A.: Semantic Segmentation of Metoceanic Processes Using SAR Observations and Deep Learning, Remote Sensing, 14, 851, https://doi.org/10.3390/rs14040851, 2022.

Ji, Y., Xu, G., Dong, C., Yang, J., and Xia, C.: Submesoscale eddies in the East China Sea detected from SAR images, Acta Oceanol. Sin., 40, 18–26, https://doi.org/10.1007/s13131-021-1714-5, 2021.

Kozlov, I. E., Artamonova, A. V., Manucharyan, G. E., and Kubryakov, A. A.: Eddies in the Western Arctic Ocean From Spaceborne SAR Observations Over Open Ocean and Marginal Ice Zones, J. Geophys. Res. Oceans, 124, 6601–6616, https://doi.org/10.1029/2019JC015113, 2019.

Stuhlmacher, A. and Gade, M.: Statistical analyses of eddies in the Western Mediterranean Sea based on Synthetic Aperture Radar imagery, Remote Sensing of Environment, 250, 112023, 2020.

Tao, M., Xu, C., Guo, L., Wang, X., and Xu, Y.: An Internal Waves Data Set From Sentinel-1 Synthetic Aperture Radar Imagery and Preliminary Detection, Earth Space Sci., 9, e2022EA002528, https://doi.org/10.1029/2022EA002528, 2022.

Topouzelis, K. and Kitsiou, D.: Detection and classification of mesoscale atmospheric phenomena above sea in SAR imagery, Remote Sensing of Environment, 160, 263–272, https://doi.org/10.1016/j.rse.2015.02.006, 2015.

Wang, C., Mouche, A., Tandeo, P., Stopa, J. E., Longépé, N., Erhard, G., Foster, R. C., Vandemark, D., and Chapron, B.: A labelled ocean SAR imagery dataset of ten geophysical phenomena from Sentinel-1 wave mode, Geosci. Data J., 6, 105–115, https://doi.org/10.1002/gdj3.73, 2019.